# Whole-Transcriptome Analysis Highlights Adenylyl Cyclase Toxins-Derived Modulation of NF-κB and ERK1/2 Pathways in Macrophages

**DOI:** 10.3390/toxins15020139

**Published:** 2023-02-09

**Authors:** Taoran Zhao, Ruihua Li, Mengyin Qian, Meirong Wang, Xiaozheng Zhang, Yuhan Wang, Xinghui Zhao, Jun Xie

**Affiliations:** 1Laboratory of Vaccine and Antibody Engineering, Beijing Institute of Biotechnology, Beijing 100071, China; 2Shanxi Key Laboratory of Birth Defect and Cell Regeneration, Department of Biochemistry and Molecular Biology, Shanxi Medical University, Taiyuan 030001, China; 3School of Chemistry and Chemical Engineering, Queen’s University Belfast, Belfast BT9 5AG, UK

**Keywords:** *Bacillus anthracis*, edema toxin, macrophage, adenylyl cyclase, cAMP, TNF-α, NF-κB, ERK

## Abstract

Edema toxin (ET), one of the main toxic factors of *Bacillus anthracis* (*B. anthracis*), is a kind of potent adenylate cyclase (AC). *B. anthracis* has adapted to resist macrophage microbicidal mechanisms in part by secreting ET. To date, there is limited information on the pathogenic mechanisms used by ET to manipulate macrophage function, especially at the transcriptome level. We used RNA sequencing to study transcriptional changes in RAW264.7 cells treated with ET. We aimed to identify molecular events associated with the establishment of infection and followed changes in cellular proteins. Our results indicate that ET inhibited TNF-α expression in the RAW264.7 mouse macrophage cell line by activating the cAMP/PKA pathway. ET challenge of macrophages induced a differential expression of genes that participate in multiple macrophage effector functions such as cytokine production, cell adhesion, and the inflammatory response. Furthermore, ET influenced the expression of components of the ERK1/2, as well as the NF-αB signaling pathways. We also showed that ET treatments inhibit the phosphorylation of the ERK1/2 protein. ET also attenuated NF-αB subunit p65 phosphorylation and transcriptional activity of NF-αB via the cAMP/PKA pathway in macrophages. Since the observed modulatory effects were characteristic only of the bacterial exotoxin ET, we propose this may be a mechanism used by B. anthracis to manipulate macrophages and establish systemic infection.

## 1. Introduction

*Bacillus anthracis (B. anthracis)* is the pathogen that causes anthrax, an acute rapidly progressing infectious disease that affects both humans and animals. The virulence factors of *B. anthracis* primarily include capsule and anthrax toxins [1]. Anthrax toxins have three components: the protective antigen (PA), the lethal factor (LF), and the edema factor (EF) [2]. LF and PA together constitute the lethal toxin (LT), which cleaves mitogen-activated protein kinase kinases (MAPKK) 1–4, 6, and 7 to inactivate the associated pathway. The edema toxin (ET) is composed of EF and PA, and EF is a calmodulin-dependent adenylate cyclase (AC). The entry of EF into host cells is mediated by the interaction between PA and host cell surface receptors; once EF enters the host cell, intracellular cyclic adenosine monophosphate (cAMP) levels increase dramatically [3].

Several studies have indicated that ET helps the early dissemination of *B. anthracis* within the host by altering the antimicrobial function of macrophages. ET markedly modified the patterns of bacterial dissemination, by leading to apparent direct dissemination to the spleen and by provoking lymphoid cell apoptosis [4]. Macrophages are critical for the early host defense response to *B. anthracis*. Previous studies have shown that mice in which macrophages were depleted were killed more rapidly by *B. anthracis* than untreated mice [5,6]. Interestingly, mice lacking the myeloid-specific toxin receptor were completely resistant to *B. anthracis* infection, while wild-type mice were highly sensitive [7]. However, the mechanisms underlying how ET affects the function of macrophages are not well studied.

cAMP, an intracellular second messenger, regulates cellular functions by its interactions with effector molecules, protein kinase A (PKA), or exchange proteins directly activated by cAMP (Epac) [8,9]. In the innate immune system, the elevation of the level of cAMP within phagocytes (including monocytes, macrophages, and neutrophils) could modulate three key effector functions of these cells: generation of inflammatory mediators (e.g., cytokine, chemokine, and lipids), phagocytosis, and intracellular killing of ingested pathogens [10]. In mammalian cells, cAMP can be synthesized by endogenous AC and degraded by phosphodiesterase (PDE) [10]. To date, there are 10 known isoforms of AC [11] and 11 distinct PDE gene families [12]. Moreover, both ACs and PDEs are differentially expressed in various cell types and are localized in different spatial compartments within the cell. As a result, cAMP signaling is under precise spatiotemporal control [13]. As an exogenous AC for the mammalian host, ET interferes with the physiological homeostasis of intracellular cAMP and down-regulates the defense function of macrophages.

EF(H351A) is an EF mutant with decreased AC activity characterized by the substitution of histidine (H) at position 351 (H351) by alanine (A) [14]. A previous study suggested that EF(H351A) represents a potential anthrax toxin decoy because it retains PA-binding ability but has significantly weaker activity [15]. However, our study found that ET(H351A) (composed of EF(H351A) and PA) can still slightly increase intracellular cAMP levels and leads to systemic toxicity in a mouse model [16]. Whether ET(H351A) could regulate macrophage function is still unknown.

Few sequencing studies have examined transcriptome changes in ET-challenged macrophages [17]. Further, no studies have measured changes in the macrophage transcriptome associated with ET(H351A) treatment. Herein, we used the RAW264.7 cell line, which is a monocyte-derived macrophage cell line, as an in vitro model to characterize global changes in gene expression in macrophages treated with ET or ET(H351A). Whole-transcriptome analysis by RNA-based next-generation sequencing (RNA-seq) shows that challenge by both ET and ET(H351A) alters the macrophage transcriptome by inducing significant changes in the expression of genes involved in various innate immune effector functions. One of the findings of our RNA-seq screen was that ET and ET(H351A) challenge influenced the expression of components in both the extracellular signal-regulated kinases 1 (ERK1) and ERK2 as well as the nuclear factor kappa B (NF-κB) signaling pathways. Further experimental verification showed a reduction in phosphorylation on the effector protein mediated the ET and ET(H351A) inhibition of ERK1 and ERK2 and NF-κB signaling pathways.

## 2. Results

### 2.1. ET and ET(H351A) Inhibited TNF-α Expression by Activating cAMP/PKA Pathway

TNF-α is a dominant factor in the macrophage response to bacterial pathogens; its level was reported to be restricted by ET [18]. After being treated with 100 ng/mL ET, the intracellular levels of cAMP in RAW264.7 cells increased over 150-fold, while treatment with 100 ng/mL ET(H531A) only induced a 3-fold increase in the intracellular cAMP levels (Figure 1A). TNF-α secretion from macrophages markedly increased upon stimulation by bacteria or pathogen-associated molecular patterns (PAMPs) (i.e., LPS, Figure 1B). This effect could be dramatically reversed by the addition of ET but not ET(H531A) (Figure 1B). Co-incubation of different concentrations of 8-Bromo-cAMP also inhibited the induction of TNF-α secretion by LPS in a dose-dependent manner (Figure 1B). With regard to the effects on transcription, LPS-induced TNF-α mRNA expression was suppressed by ET and 8-Bromo-cAMP, as well as by ET(H351A) (Figure 1C). However, in the absence of LPS, ET(H351A) and 8-Bromo-cAMP showed weaker inhibitory effects on the transcription of TNF-α (Figure 1D). Together, these results indicated that ET inhibits LPS-induced TNF-α expression in macrophages by elevating intracellular cAMP levels.

We next investigated whether ET inhibits TNF-α expression in macrophages by activating PKA, a main downstream target of intracellular cAMP [9]. H89 is a widely used PKA inhibitor. In the presence of LPS, both ET-mediated and cAMP-mediated inhibition of TNF-α secretion was reversed by H89 (Figure 1E). Furthermore, we transfected the luciferase reporter gene for the TNF promoter into RAW264.7 cells to determine the effects of ET and H89 on TNF promoter activation. ET alone could inhibit TNF promoter activation, while H89 alone did not have a significant effect (Figure 1F). When H89 was applied to macrophages with ET, ET-induced inhibition of the TNF promoter was reversed (Figure 1F). Thus, activation of PKA is involved in the ET-mediated decrease in TNF-α secretion.

### 2.2. ET and ET(H351A) Induced Global Changes in Gene Expression of Macrophages

To assess the changes in gene expression after ET challenge, whole-transcriptome analysis was performed by RNA-based next-generation sequencing (RNA-seq) using RAW264.7 mouse macrophages, which were challenged with PA, ET(H351A), or ET. Figure 2A shows the sample correlation/clustering study of gene expression profiles, which clearly showed a distinct pattern of the total RNA of ET-treated samples vs. ET(H351A)-treated samples. The two samples clustered for each experimental condition, showing that sample variability was not a major contributor to our data set. Interestingly, the transcriptional profile of ET-stimulated macrophages clearly separated from the control group (PA treated), while ET(H351A)-treated samples clustered together with PA-treated samples (Figure 2A). 

Next, differential expression analysis was carried out between each treatment condition using the limma method, where the standard of *p*-value cutoff ≤ 0.05 and the log fold change |log_2_FC| ≥ 1 was utilized to compile a list of differentially expressed genes (DEGs) for further analyses. The degree of DEGs was determined by the treatments and was plotted based on whether genes were up- or down-regulated compared to the PA group (Figure 2B). Stimulation of macrophages with ET induced changes in 4094 genes: 2046 (49.98%) and 2048 (50.02%) genes were up- and down-regulated, respectively. It should be noted that ET(H351A) stimulation induced 1107 DEGs, which was fewer than the number of DEGs obtained following ET treatment (Figure 2C).

The most significant DEG for each treatment was identified upon inspection of Figure 2D. Of these, Ptchd1 and Scara3 were down-regulated while Thbs1 and Rab44 were up-regulated in both ET- and ET(H351A)-treated samples compared to PA-treated samples. The Ptchd1 gene encodes a protein involved in synaptic transmission, whose deficiency induces a neurodevelopmental disorder [19,20]. The Scara3 gene encodes a macrophage scavenger receptor-like protein and was reported to protect cells from oxidative stress-induced cell damage by removing oxidizing molecules or harmful products of oxidation [21,22]. The Thbs1 gene encodes Thrombospondin-1, associated with platelet activation and wound healing [23]. Rab44 gene levels are commonly decreased in macrophages during differentiation from their precursor cells; however, short-term treatment with IFN and LPS could elevate the level of Rab44 in macrophages [24].

In addition, there are 693 common DEGs among all DEGs induced by ET or ET(H351A) treatment. Hierarchical clustering analysis showed that these common DEGs were classified into three clusters (Figure 2E). Cluster 1 included approximately half of the common DEGs that exhibited up-regulated expression in the ET or ET(H351A) treatment groups relative to the PA treatment groups. However, the DEGs in cluster 1 presented higher levels of increase in the ET treatment groups than those in the ET(H351A) groups. By contrast, the rest of the DEGs in clusters 2 and 3 displayed lower levels of down-regulation in the ET treatment groups than those in the ET(H351A) groups.

### 2.3. ET and ET(H351A) Influenced Macrophage Biological Processes

To better understand and classify the biological implications of DEGs induced by ET or ET(H351A) stimulation in macrophages, the enrichment of DEGs in the gene ontology (GO) category of biological process was analyzed using the ClusterProfiler tool. Overall, 307 and 77 significant (adjusted *p*-value < 0.01) biological processes were identified for ET and ET(H351A) treatment, respectively (Appendix A). To analyze the relationship between the enriched terms, as shown in Appendix A, the most significant GO terms were structured in the form of a directed acyclic graph (DAG) to represent a network of complex correlation of ‘child’ and ‘parent’. The more ‘child’ a GO term is, the more the term is related to a specific biological process. Figure 3A reports that the most ‘child’ terms in the ET(H351A) treatment group included cell chemotaxis, leukocyte cell–cell adhesion, inflammatory response, positive regulation of cytokine production, positive regulation of peptidyl-tyrosine phosphorylation, regulation of cell activation, and response to bacterium and T cell activation. The most ‘child’ terms in the ET treatment group included those in the positive regulation of apoptotic processes, positive regulation of cell adhesion, regulation of the mitogen-activated protein kinase (MAPK) cascade, regulation of ERK1 and ERK2 cascades, and rRNA processing. It is worth noting that the DEGs involving rRNA processing were all down-regulated by ET, while DEGs involving the other four biological processes were consistently up- and down-regulated (Figure 3B). 

The enrichment of common DEGs between the ET or ET(H351A) treatment groups was also analyzed. Figure 3C reports that the most significantly enriched biological processes were the regulation of the ERK1 and ERK2 cascade, regulation of epithelial cell proliferation, positive regulation of cytokine production, cellular response to biotic stimulus, and cell chemotaxis. Most of these processes are related to the innate immune response of the macrophage.

To further analyze the similarities and differences between ET and ET(H351A) treatments in affecting the macrophage biological processes, the count of DEGs induced by each treatment and the corresponding *p*-value of each biological process above were plotted (Figure 3D). In the ET(H351A) treatment group, regardless of rRNA processing, the remaining 14 biological processes were all significantly enriched (adjusted *p*-value < 0.01) with 20 to 60 DEGs in each process. However, in the ET treatment group, 12 of 15 biological processes were significantly enriched, with many more DEGs in each process (70 to 160 DEGs). 

From the enrichment results of biological processes in the GO analysis, DEGs induced by ET or ET(H351A) treatment were significantly enriched in the regulation of the ERK1 and ERK2 cascade (ID: GO 0070372) (Figure 3C,D). A total of 83 and 29 DEGs in the ET and ET(H351A) treatment groups, respectively, were enriched in this cascade and the FPKMs of each DEG are shown in a heat map (Figure 4A, Appendix A). Several cytokine or cytokine-related genes were involved in the regulation of the ERK1 and ERK2 cascade, including CCL2 and IL-6. However, the key elements ERK1 and ERK2 were not affected by ET or ET(H351A) treatment at the mRNA level. ERK1/2 can be activated and phosphorylated under LPS stimulation. PD0325901 is a potent ERK1/2 phosphorylation inhibitor. The total and phosphorylated ERK1/2 levels were evaluated by Western blotting. Densitometry analysis of Western blot bands showed that ET stimulation down-regulated both total and phosphorylated ERK1/2 levels, while ET(H351A) treatment alone showed a down-regulation of total ERK1/2 (Figure 4B–D).

### 2.4. ET and ET(H351A) Modulated Cytokine Pathways and Signaling Pathways

Next, we identified pathways relevant to the challenge with ET or ET(H351A). Using ClusterProfiler, the list of DEGs was mapped onto predefined pathways from the Kyoto Encyclopedia of Genes and Genomes (KEGG) database. We limited our analysis to highly significant pathways with a *p* < 0.01, which resulted in 25 and 17 pathways for ET and ET(H351A) treatment, respectively (Table A1). The ET(H351A)/ET-macrophage transcriptome reinforced the pathogenic potential of AC toxins by the number of significant pathways linked to pathogens that subvert immune cells (malaria, Epstein–Barr virus, legionellosis, herpesvirus, African trypanosomiasis). Meanwhile, the type I diabetes mellitus pathway was significantly affected in the ET(H351A) treatment group, and the pathways of cancer, rheumatoid arthritis, and inflammatory bowel disease in both the ET(H351A) and ET treatment groups. Macrophages, and especially the intracellular cAMP levels, continue to be linked to major diseases like those listed above [25], and these results suggest that regulation of intracellular cAMP levels could either play a role in the pathogenesis of these diseases or may represent a potential treatment approach. Moreover, several pathways involved in cell proliferation (ribosome biogenesis in eukaryotes, aminoacyl-tRNA biosynthesis, pyrimidine metabolism, one carbon pool by folate) were just enriched in the ET-treated group but not the ET(H351A)-treated group.

Cytokine secretion is an important means for macrophages to inhibit pathogen invasion. In both the ET(H351A) and ET treatment groups, the cytokine–cytokine receptor interaction pathway enriched most DEGs by KEGG analysis (Figure 5A,B). In the RNAseq data, expression of the cytokine genes TNF and CCL2 was significantly down-regulated, and IL-6 was up-regulated in both the ET(H351A) and ET treatment groups (Figure 6A). We then investigated the expression of these cytokines at the protein level in response to different treatments. LPS-stimulated macrophages produced higher levels of TNF-α and CCL2, along with a lower level of IL-6 (Figure 6B). Consistent with RNAseq results, macrophages co-treated with ET and LPS synthesized lower levels of the pro-inflammatory cytokines TNF-α and CCL2, but higher levels of IL-6 (Figure 6B). The effects of ET(H351A) were similar to those of ET but much weaker. ET(H351A) limited LPS-induced TNF-α and CCL2 production but increased IL-6 secretion (Figure 6B). Interestingly, ET treatment also induced a significant increase in IL-10 secretion in macrophages with LPS stimuli (Figure 6B). However, IL-10 gene expression did not show any significant difference in the RNAseq data.

Furthermore, the NF-κB signaling pathway was significantly affected in both the ET(H351A) and ET treatment groups (Figure 5A,B), with 13 and 34 DEGs enriched in each group, respectively. The enriched DEGs are shown in a heatmap (Figure 6C, Appendix A) to further clarify the component regulated by ET(H351A) or ET. The NF-κB signaling pathway, and two cytokine genes Ccl4 and TNF were both down-regulated by ET(H351A) and ET treatments, but the cytokine gene IL-1b was up-regulated by ET(H351A) and ET, while two NF-κB signaling module component genes Relb and Iκbκb were up-regulated by ET, but not by ET(H351A). 

### 2.5. ET Down-Regulated NF-κB Transcription Activity and p65 Phosphorylation

To verify the regulation of the NF-κB pathway in response to ET or ET(H351A) stimulation, the luciferase reporter vector for NF-κB binding sites was transfected into RAW264.7 cells to reflect the binding potential of NF-κB to its target genes. The binding activity of NF-κB was significantly activated by LPS stimulation, while BAY 11-7082 (a specific NF-κB inhibitor) could abolish the activation due to LPS (Figure 7A). Interestingly, this activation effect of LPS could also be partially inhibited by ET or ET(H351A) (relative to the suppression effect of BAY 11-7082) (Figure 7A). Furthermore, in the absence of LPS, ET, but not ET(H351A), also partially suppressed the binding activity of NF-κB (Figure 7B). We next investigated whether ET inhibited NF-κB signaling by activating PKA, a main downstream target of intracellular cAMP. When H89 was applied to macrophages with ET, the ET-induced NF-κB binding inhibition was reversed (Figure 7C). Together, this data indicated that ET inhibits NF-κB signaling by activating the cAMP/PKA pathway.

To determine whether the observed inhibition of NF-κB signaling was due to the increase in the levels of the inhibitor of kappa Bα (IκBα), the expression level of IκBα was evaluated by Western blotting. Modest differences in IκBα expression between cells treated with ET or ET(H351A) and untreated cells were identified (Figure 7D). The abundance of p65, the effector subunit of NF-κB, was further measured in RAW264.7 cells. Densitometry analysis of the Western blot bands for total p65 showed that stimulation with ET or ET(H351A) had no effect on p65 abundance (Figure 7D). However, treatment with both ET and BAY heavily impaired the phosphorylation of a key amino acid residue of p65 (S536) (Figure 7E). These data show that ET partially modulates the activation of the NF-κB signaling pathway by selectively interfering with the phosphorylation of p65.

## 3. Discussion

Macrophages, which are a dynamic and heterogeneous cell type, are well known as an important component of innate host antibacterial immunity [26]. Macrophages are roughly classified into two groups: classically activated macrophages (M1) and alternatively activated macrophages (M2). The M1 phenotype macrophages produce high levels of pro-inflammatory cytokines, including TNF-α, CCL2, IL-1b, and IL-6, to kill microorganisms and increase the Th1 immune response [26,27]. By contrast, M2 phenotype macrophages are characterized by the low production of the pro-inflammatory cytokines and high production of the anti-inflammatory cytokine IL-10 [26,27].

Several bacterial pathogens have evolved strategies to interfere with macrophage activation and to modulate host responses [28]. For example, Mycobacterium tuberculosis induces the polarization of macrophages toward the M1 phenotype during the early stages of infection [29,30], but polarizes the macrophages to the M2 phenotype at a later stage of infection via the virulence factor early secretory antigenic target ESAT-6 [31,32]. Coxiella burnetii, an obligate intracellular bacterium, survives in macrophages by stimulating an atypical M2 activation program [33]. Interestingly, certain bacteria have evolved to hijack the host cAMP axis by increasing the intracellular cAMP production of the host cell [34]. For instance, the pertussis toxin and CyaA of Bordetella pertussis, and the cholera toxin of Vibrio cholera, have both been reported to inhibit the host defense functions of myeloid phagocytes [34].

In this study, we demonstrated that ET and ET(H531A) treatment increased macrophage intracellular cAMP concentration and reduced macrophage TNF-α expression. Furthermore, TNF-α expression was negatively related to intracellular cAMP in a dose-dependent manner. ET inhibited TNF-α expression through the cAMP/PKA pathway. Furthermore, ET-treated macrophages produced higher anti-inflammatory cytokine IL-10 levels. Although a switch from M1 to M2 macrophage polarization occurs under various physiological and pathological conditions, TNF-α has been identified as a major anti-M2 factor [27]. In fact, many studies have shown an inverse relationship between the degree of TNF-α signaling and the number of M2 macrophages [27]. It is possible that in sepsis, TNF-α production could precede macrophage expansion [35], and monocytes could be induced into a pro-inflammation M1 phenotype by TNF-α stimulation [36]. Convincingly, the complete knockout of TNF led mice to increase the expression of M2-linked genes and M2 macrophage expansion [37]. Therefore, the suppressive effect of ET on TNF-α production might be the key to the induction of the M2 macrophage phenotype.

However, IL-6, a cytokine typically associated with M1 polarization, was increased in ET-treated macrophages. This is consistent with previous studies on the relationship between macrophage polarization and cAMP signaling [38] or the Q fever pathogen *C. burnetii* [34]. In these studies, both stimuli inhibit TNF-α while inducing IL-6 in macrophages. Studies have indicated that IL-6 may inhibit the IFN-γ response during M. tuberculosis infection [39] and reduce the Th1 response in Yersinia enterocolitica-infected mice [40]. IL-6 and TGF-β1 act together to induce IL-10 production in T cells [41]. Thus, it is tempting to speculate that IL-6 may contribute to the immune modulatory role of macrophages. According to the review by Abbas et al., M2 macrophages can be further divided into several subsets [42], and ET or other cAMP agonists may induce macrophages into atypical M2 subsets. 

In 2006, Comer et al. performed a chip analysis on ET-treated mouse macrophages, and the results showed that ET treatment for 3 h and 6 h changed the expression levels of 71 and 259 genes, respectively [17]. Although these differentially expressed genes have shown that ET has strong and extensive cellular activity, relative to the overall transcriptome, these genes may not reflect the full impact of ET. In our study, ET treatment for 8 h induced 4094 DEGs, reflecting a widespread influence of ET on macrophages. Even ET(H531A), once thought to be a non-toxic mutant, induced 1107 DEGs in 8 h. This indicates that the use of ET(H531A) for the treatment of anthrax infection should be carefully re-considered. 

cAMP not only has a comprehensive immune-cell regulatory function but also participates in the activity and development of the nervous system [43]. The cAMP/PKA signaling pathway is critical for long-lasting synaptic and memory formation [44]. Disruption of this pathway by certain toxins could result in neurodevelopmental damage [45,46]. Ptchd1 is among the genes most negatively regulated by ET or ET(H351A) treatment. Ptchd1 encodes a transmembrane protein, whose mutation or deficiency is involved in neurodevelopmental disorders [19,20]. Therefore, inhibition of the Ptchd1 signal may be a mechanism of the cAMP/PKA pathway that contributes to neurodevelopmental defects.

In eukaryotes, cAMP synthesis is canonically triggered via G protein-coupled receptor (GPCR)-mediated activation of endogenous transmembrane ACs. The functional diversity of cAMP signaling is tightly regulated by intracellular cAMP gradients and microdomains [47]. The destruction of cAMP compartmentation in normal cells increases cell proliferation and induces cell transformation [48]. ET, as an exogenous AC, intensively elevates the cAMP level in macrophages independent of GPCR. The cAMP molecules induced by ET are very likely distributed randomly within the cell. That may be the reason why ET treatment affected several cell proliferation-related pathways.

An interesting observation from the RNAseq analysis was that ET and ET(H351A) modulated the MAPK signaling pathway (especially the ERK1/2 cascade) and the NF-κB signaling pathway. The innate immune response provides the first line of defense after infection, using a limited number of germline-encoded pattern recognition receptors (PRRs) to recognize the PAMPs of invariant pathogens [49,50]. Macrophages express several classes of PRRs, including Toll-like receptors (TLRs), RIG-I-like receptors (RLRs), NOD-like receptors (NLRs), and C-type lectin receptors (CLRs). Although receptor-proximal signaling mechanisms vary, all of these PRRs activate MAPK and NF-κB pathways, which are crucial for generating immune responses [49,50].

In the present study, the KEGG analysis of DEGs in macrophages induced by ET or ET(H351A) treatment suggested the enrichment of the NF-κB signaling pathway. NF-κB signaling is a master regulator of immunological transcriptional programs, including the inflammatory response to pathogens by innate immune cells [51]. Interactions between cAMP and NF-κB cascades have been described in various cell types, including, among others, diverse leukocyte subsets, fibroblasts, epithelial and endothelial cells, smooth muscle cells, and brain cells. Some studies have reported cell-type-specific effects of cAMP. For instance, cAMP inhibited NF-κB in 3T3 fibroblasts, whereas it induced NF-κB in brown adipocytes [52]. The NF-κB signaling module consists of five NF-κB monomers (RelA/p65, RelB, cRel, NF-κB 1 p50, and NF-κB p52), which can act as homo- or heterodimers, and five inhibitory proteins (IκB α, β, ε, γ, and δ) that make up the IκB protein family. The inactivated NF-κB proteins are sequestered in the cytoplasm through physical interaction with IκB proteins. Upon bacterial infection, the PAMPs of pathogens activate NF-κB signaling via the activation of the inhibitor kappa B kinase (IKK) trimeric complex. Once activated by phosphorylation, IKK further phosphorylates the IκB, which leads to the degradation of IκB and the release of NF-κB from the NF-κB/IκB complex, which allows NF-κB to activate the transcriptional activity of its target genes [52].

The p50/p65 combination is the most abundant and ubiquitously expressed NF-κB heterodimer. In the present study, ET treatment decreased the phosphorylation of the p65 subunit of the NF-κB transcription factor. This impairment may have led to the failure of this heterodimer to enter the nucleus. Several sites in the human and murine TNF promoters are designated as κB motifs, and these motifs are involved in the NF-κB -mediated regulation of TNF-α transcription [53]. Thus, ET interfered with the binding of NF-κB to its target genes (including TNF-α).

There are two main types of intracellular cAMP transducers: cAMP-dependent PKA and the guanine exchange proteins that are directly activated by cAMP (EPAC-1 and EPAC-2). Of these, PKA is considered the main effector of cAMP in interacting with NF-κB [52]. The specific PKA inhibitor H89 reversed the ET-induced inhibition of NF-κB, suggesting the involvement of the cAMP/PKA pathway. The S536 residue of p65 can be phosphorylated by several kinases, including IKKα, IKKβ, Akt, TANK-binding kinase1 (TBK1), IKKε, and so on [54]. Among these kinases, the activity of non-canonical IκB kinases TBK1 and IKKε has been reported to be inhibited by cAMP increasement and PKA activation [55]. This may be the possible mechanism by which ET inhibits NF-κB transcription activity.

MAKP includes four subsets: ERK1 and ERK2 (P44MAPK and P42MAPK, respectively); stress-activated protein kinases (SAPKs/JNKs); p38 kinase; and ERK5. The ERK1 and ERK2 pathways have been shown to have important roles in macrophages, regulating cytokine production via both transcriptional and post-transcriptional mechanisms. This study has shown that ET and ET(H351A) diminished the phosphorylation of ERK1 and ERK 2 as well as down-regulated the total protein levels of ERK1 and ERK 2. Activation of ERK1 and ERK2 signaling by all TLRs in primary macrophages is mediated by the MAP3K TPL2 [56]. In unstimulated cells, TPL2 forms a complex with the NF-κB subunit precursor protein p105, which inhibits the kinase activity of TPL2 [57,58]. TLR stimulation activates the IKK complex, which phosphorylates p105, inducing its K48-linked ubiquitylation and proteasome-mediated proteolysis [28]. After its release from p105-mediated inhibition, TPL2 can then phosphorylate MAPK kinase 1 (MKK1) and MKK2 upstream of ERK1 and ERK2. IKK2 also directly phosphorylates TPL2 at Ser400, which is a crucial regulatory residue in its carboxyl terminus that is required for LPS to induce ERK activation in macrophages [59,60]. This crosstalk between the ERK1 and ERK2 pathways and the NF-κB pathway may explain why these signaling pathways are always regulated simultaneously in macrophages.

## 4. Conclusions

In this study, we show that both ET(H351A) and ET induce significant changes in the macrophage transcriptome. In silico analysis demonstrated that the biological processes involved were the regulation of the ERK1 and ERK2 cascade, regulation of epithelial cell proliferation, positive regulation of cytokine production, cellular response to biotic stimulus, and cell chemotaxis. Moreover, ET(H351A) and ET modulated both the cytokine-related pathways and NF-κB signaling pathways. Further experimental verification suggested that the inhibition of ERK1 and ERK2 of phosphorylation, as well as p65, may be the main targets of ET(H351A)- and ET-mediated modulation of the ERK1 and ERK2 as well as NF-κB signaling pathways. Our study provides novel insight into how the AC toxin helps pathogens evade host defense mechanisms, and may serve as a framework for further studies of *B. anthracis* infection prevention and treatment.

## 5. Materials and Methods

### 5.1. Toxins

The PA, EF, and EF(H351A) proteins used in this investigation were expressed in *Escherichia coli* and were purified as previously described [16,61]. EF(H351A) is a variant of EF with a mutation of histidine (H) 351 into alanine (A) that leads to a mostly diminished but not eliminated AC activity [16]. The treatment of 100 ng/mL ET in this study suggests the combination of 100 ng/mL EF with 200 ng/mL PA, just as 100 ng/mL ET(H351A) indicates the co-treatment of 100 ng/mL EF(H351A) with 200 ng/mL PA.

### 5.2. Cell Culture

The monocyte-derived mouse macrophage cell line RAW264.7 was obtained from the American Type Culture Collection and cultured in minimum essential medium (MEM) supplemented with 10% fetal bovine serum, penicillin (100 U/mL), streptomycin (100 μg/mL), and glutamine (2 mM) at 37 °C under 5% CO_2_. Before stimulation, RAW264.7 cells were seeded in 6-well plates at a density of 4 × 10^5^ cells/well and cultured overnight.

### 5.3. Intracellular cAMP Measurement

RAW264.7 cells pretreated with 200 ng/mL PA, 100 ng/ mL ET(H351A), or 100 ng/mL ET for 8 h were lysed using 0.1 M HCI. Total intracellular cAMP levels were assayed using the Monoclonal Anti-cAMP Antibody Based Direct cAMP ELISA Kit (Neweast Bioscience, Wuhan, China), following the manufacturer’s instructions.

### 5.4. Cytokine Production

Supernatants were collected from all RAW264.7 cells in all pretreatment groups (10 ng/mL LPS (Sigma-Aldrich, Steinheim, Germany) plus 200 ng/mL PA, 100 ng/mL ET(H351A), or 100 ng/mL ET) and the control group. The levels of tumor necrosis factor-α (ΤNF-α), C-C motif chemokine 2 (CCL2), Interleukin-6 (IL-6), and IL-10 in the supernatants were determined using the cytometric bead array (CBA) mouse inflammation kits (BD Biosciences, San Jose, USA), following the manufacturer’s instructions.

### 5.5. Gene Expression Analysis with Quantitative Reverse Transcriptase-PCR

Total RNA was extracted using RNeasy Plus Mini Kits (Qiagen, Duesseldorf, Germany) and reversed transcribed to cDNA using QuantiTect Reverse Transcription Kits (Qiagen, Duesseldorf, Germany), following the manufacturer’s instructions. A LightCycler (ABI Prism 7000) and an SYBR RT-PCR kit (Takara, Tokyo, Japan) were used for quantitative reverse transcriptase-PCR (qRT-PCR) analysis. TNF was amplified using the specific primer pair 5′-GGTCTGGGCCATAGAACTGA-3′ and 5′-CAGCCTCTTCTCATTCCTGC-3′, while β-actin was amplified using the specific primer pair 5′-ATGGAGGGGAATACAGCCC-3′ and 5′-TTCTTTGCAGCTCCTTCGTT-3′. The expression of TNF in each sample was normalized to β-actin expression.

### 5.6. Luciferase Reporter Gene Expression Assay

RAW264.7 cells (2 × 10^5^ cells/well) were seeded in 24-well plates. After 12 h, cells were co-transfected with 0.25 μg of Renilla-expressing plasmids (pRL-SV40-C; Beyotime Biotechnology, Shanghai, China) and either 1 μg of NF-κB binding site reporter plasmids (pNF-κB-TA-Luc; Beyotime Biotechnology, Shanghai, China) or 1 μg of TNF-α promoter reporter plasmids (pTNF-α-promoter-Luc; Beyotime Biotechnology, Shanghai, China) using TurboFect (Invitrogen, Carlsbad, USA). At 4 h post-transfection, cells were treated with PBS, 100 ng/mL ET(H351A), 100 ng/mL ET, 10 μM BAY 11-7082 (BAY), and 40 μM H89, in the presence or absence of 10 ng/mL LPS or 10 ng/mL TNF-α. Luciferase activity levels were determined using the Dual-Luciferase reporter assay system (Promega, Madison, WI, USA) using a microplate luminometer (GLOMAX96; Promega, Madison, WI, USA), following the manufacturer’s instructions. Firefly luciferase activity was normalized against Renilla luciferase activity.

### 5.7. RNA Isolation, Sequencing, and Analysis

RAW264.7 cells were seeded in 10-cm dishes (10^7^ cells/dish) and cultured in the incubator overnight. The next morning, cells were stimulated with PA 200 ng/mL, ET(H351A) 100 ng/mL, or ET 100 ng/mL for 8 h. After stimulation, the culture medium in each plate was discarded and the cells were harvested using Trizol (Life Technologies, Carlsbad, CA, USA) and stored at −80 °C until RNA extraction. Cells suspended in Trizol were transported on dry ice to Zhongkejingyun Bio-Information Technology Co., Ltd., (Beijing, China), where total RNA was isolated and RNA quality control was conducted. After passing the quality inspection, the rRNA was removed by hybridization capture based on the structure and sequence characteristics of the rRNA, and the remaining RNA samples were used for reverse transcription and library construction. The Illumina HiSeqTM4000/MisseqTM/X-Ten high-throughput sequencing platform was used to sequence the cDNA library and the raw sequencing data were analyzed with FastQC using Cutadapt to remove joints and Trimmomatic to remove low-quality bases and reads at both ends. The clean data were aligned to the Mus musculus reference genome assembly (GRC39.fa) using Hisat2, generating alignment files in BAM format. The number of fragments that overlap each Entrez gene was summarized using featureCounts, differential expression analysis between each challenge (ET(H351A) 100 ng/mL or ET 100 ng/mL), and the control condition (PA 200 ng/mL) was performed using the limma software package. A q-value cutoff ≤ 0.05 with an absolute |log2FC| ≥ 1 was used to determine differential expression.

### 5.8. Western Blotting

The stimulated RAW264.7 cells were suspended in radioimmunoprecipitation assay (RIPA) buffer containing protease inhibitors (protease inhibitor cocktail tablets; Roche Applied Sciences, Mannheim, Germany) and phosphatase inhibitors (TransGen Biotech, Beijing, China) for 20 min. The proteins in the cell lysates were separated by electrophoresis, and the separated proteins were transferred to PVDF membranes (Millipore, MI, USA). After an overnight incubation in Tris-buffered saline supplemented with 0.2% Tween 20 (TBST) and 5% nonfat dry milk, membranes were incubated with antibodies against IκBα (1:500; Santa Cruz Biotechnology), p65 (1:500; Santa Cruz Biotechnology), p65S-phosphor-S536 (1:500, Santa Cruz Biotechnology), p-ERK1/2 (1:500, Cell Signaling Technology), ERK1/2 (1:1000, Cell Signaling Technology), or β-actin (1:1000, Abcam) for 2 h at room temperature. The membranes were washed with TBST and then incubated with the peroxidase-conjugated secondary antibodies (1:5000; Abcam) at room temperature for 1 h. ImmobilonTM Western Chemiluminescent HRP Substrate (Millipore) and the Western blotting imager (Clinx Scinence Instruments Co., Shanghai, China) were used to determine the protein expression.

### 5.9. Statistical Analysis

GraphPad Prism 7.0 was used for statistical analyses. Significant differences between the means of the experimental and control groups were identified with Student’s *t*-test or with one-way ANOVA analysis. We considered *p* < 0.05 as statistically significant.

## Figures and Tables

**Figure 1 toxins-15-00139-f001:**
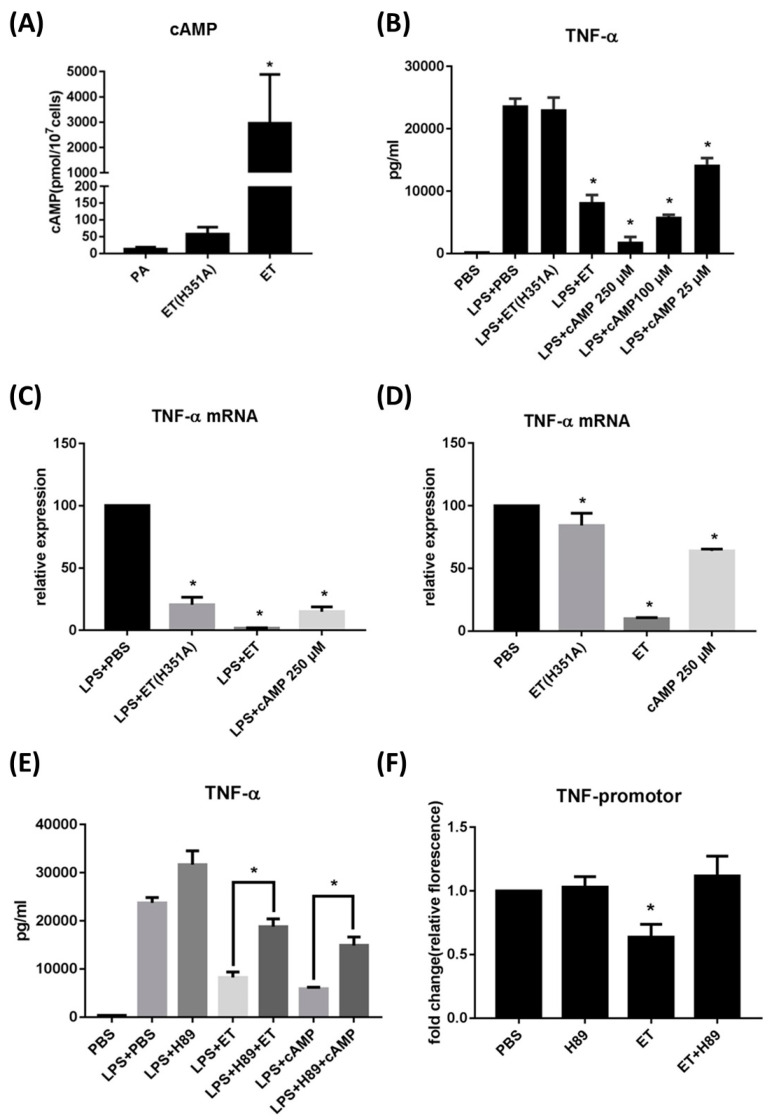
ET and ET(H351A) inhibit TNF-α expression via cAMP induction and PKA activation. (**A**) Intracellular cAMP levels in RAW264.7 cells treated with 200 ng/mL PA, 100 ng/mL ET (H351A), or 100 ng/mL ET for 3 h. (**B**) TNF-α production of RAW264.7 cells induced with PBS, 100 ng/mL ET (H351A), 100 ng/mL ET, or 25–250 μM 8-Bromo-cAMP for 3 h in the presence of LPS. The PBS-treated group acted as the blank control. * *p* < 0.05 as compared to the LPS+PBS group. (**C**,**D**) mRNA expression of TNF-α in RAW264.7 cells induced by PBS, 100 ng/mL ET (H351A), 100 ng/mL ET, or 250 μM 8-Bromo-cAMP for 3 h in the (**C**) presence or (**D**) absence of LPS. (**E**) Relative luciferase expression in RAW264.7 cells after transfection with the TNF promoter activation reporter plasmid and the control plasmid, followed by 3 h treatment with PBS, 10 μM H89, 100 ng/mL ET, or both H89 and ET. (**F**) Protein expression of TNF-α in RAW264.7 cells induced by PBS, 10 μM H89, 100 ng/mL ET, both H89 and ET, 250 μM cAMP, or both H89 and cAMP, in the presence of LPS. The PBS-treated group acted as the blank control. * *p* < 0.05 as compared to the LPS + PBS group. Results represent the mean ± SD of three independent experiments. * *p* < 0.05.

**Figure 2 toxins-15-00139-f002:**
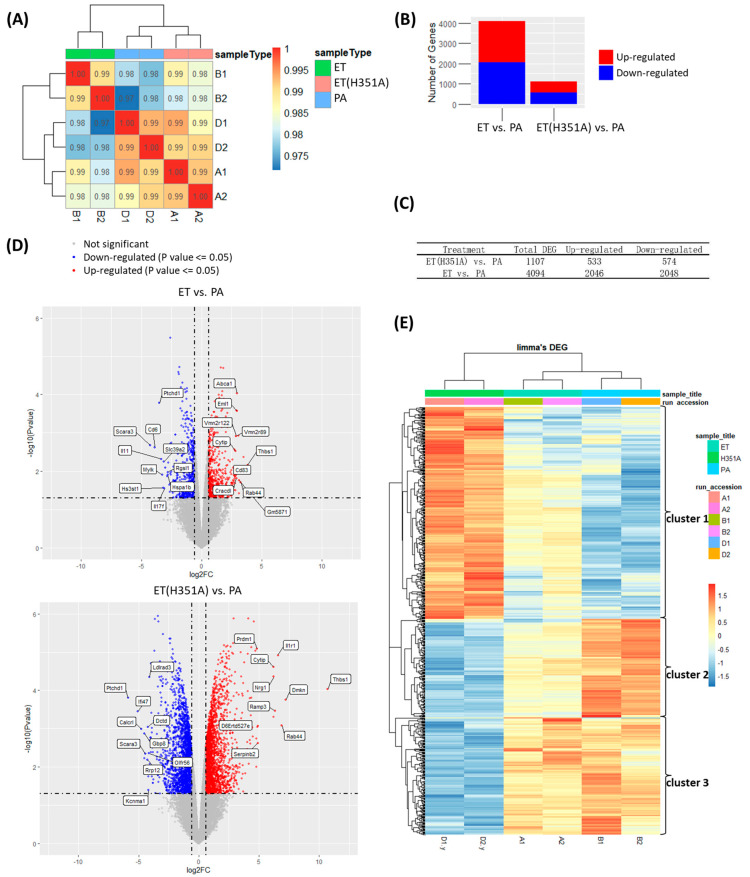
Global changes in the transcriptome of ET-, ET(H351A)-, or PA-treated macrophages. Illumina RNA sequencing was performed on mouse macrophage cells RAW264.7 treated with ET, ET (H351A), or PA for 8 h. (**A**) Heatmap showing the hierarchical clustering of sample correlations. The dendrogram clustered the significantly regulated treatment conditions with the most similar expression profiles among the 3 treatments. The correlation is represented by a color code shown in the legend where red indicates a higher correlation. The color code distinguishes the treatment conditions (ET, ET(H351A), or PA). (**B**,**C**) For each treatment, the average number of up-regulated and down-regulated differentially expressed genes (DEG) was determined and plotted. (**D**) The volcano plot of the DEGs. Genes with *p* ≤ 0.05 and log_2_ FC ≥ 1 are colored in red, genes with *p* ≤ 0.05 and log_2_ FC ≤ 1 are colored in blue, and genes with high fold changes are labeled. (**E**) Heatmap showing hierarchical clustering of the common DEGs between ET and ET (H351A) treatment groups. Orange and blue represent up-regulation and down-regulation, respectively, of the DEGs.

**Figure 3 toxins-15-00139-f003:**
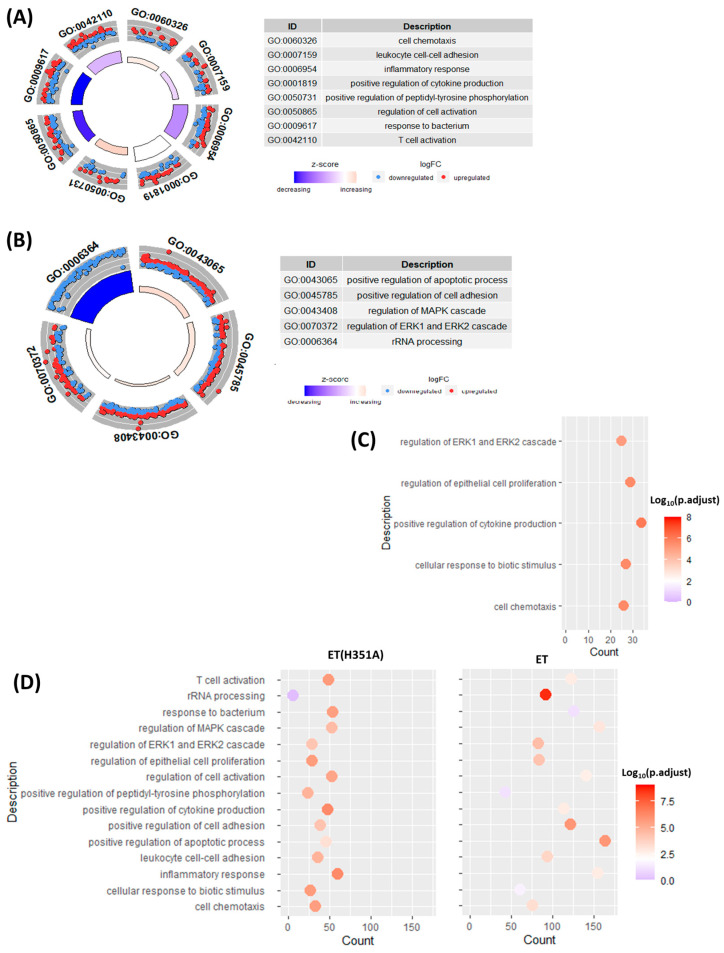
Biological processes affected during ET or ET(H351A) challenge. The list of differentially expressed genes (DEGs) (|log_2_ FC| > 1; *p* < 0.05) for each group was analyzed for biological processes enrichment using the ClusterProfiler tool. The enrichment results were visualized using GOplot and GGplot2 tools. (**A**,**B**) The most ‘child’ biological processes among the most significantly enriched GO terms in ET(H351A) or ET treatment groups. The size and color of the inner trapezoid indicate the adjusted *p*-value and stand score (z-score), respectively, of each GO term. The color and distance to the inner edge of the dots in the outer trapezoid indicate the regulated form and fold change of the involved DEGs, respectively, of each GO term. (**C**) The most significantly enriched biological processes emerging from the list of common DEGs between ET and ET(H351A) treatment groups are graphed according to the DEG count defined by each GO term. The color of each dot indicates the −log10 of the adjusted *p*-value. (**D**) The comparison of the number of DEGs and the significantly enriched biological processes between the ET and ET(H351A) treatment groups. The color of each dot indicates the −log10 of the adjusted *p*-value.

**Figure 4 toxins-15-00139-f004:**
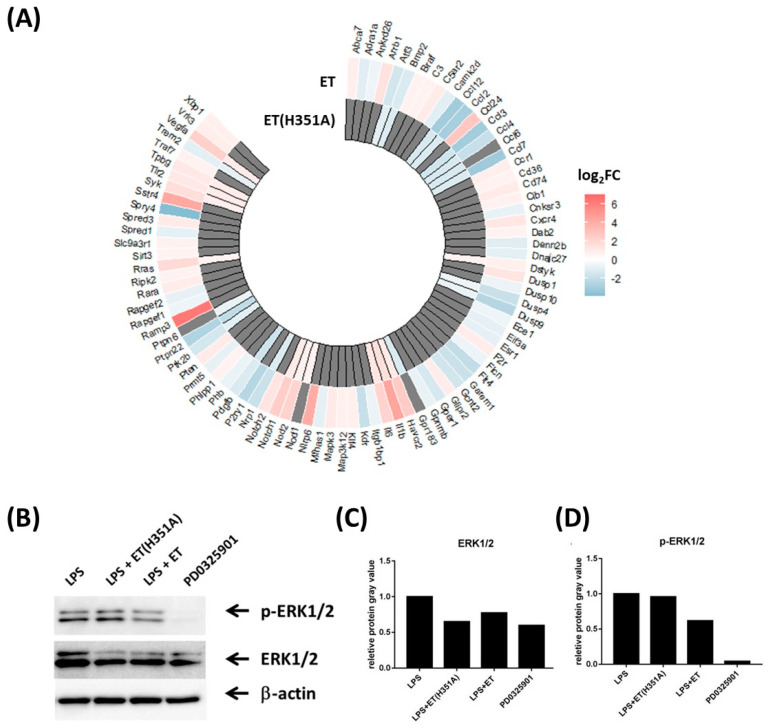
ET and ET(H351A) effects on the ERK1 and ERK2 cascade. (**A**) Heatmap of the DEGs in the regulation of the ERK1 and ERK2 cascade (ID: GO 0070372). Red and blue represent up-regulation and down-regulation, respectively, of the mRNAs. (**B**) Total and phosphorylated ERK1/2 in RWA264.7 cells treated with LPS, LPS + ET (H351A), LPS + ET, or PD0325901 for 3 h. Shown are the representative Western blots, and the relative protein expression was normalized against β-actin and quantified by densitometry (**C**,**D**).

**Figure 5 toxins-15-00139-f005:**
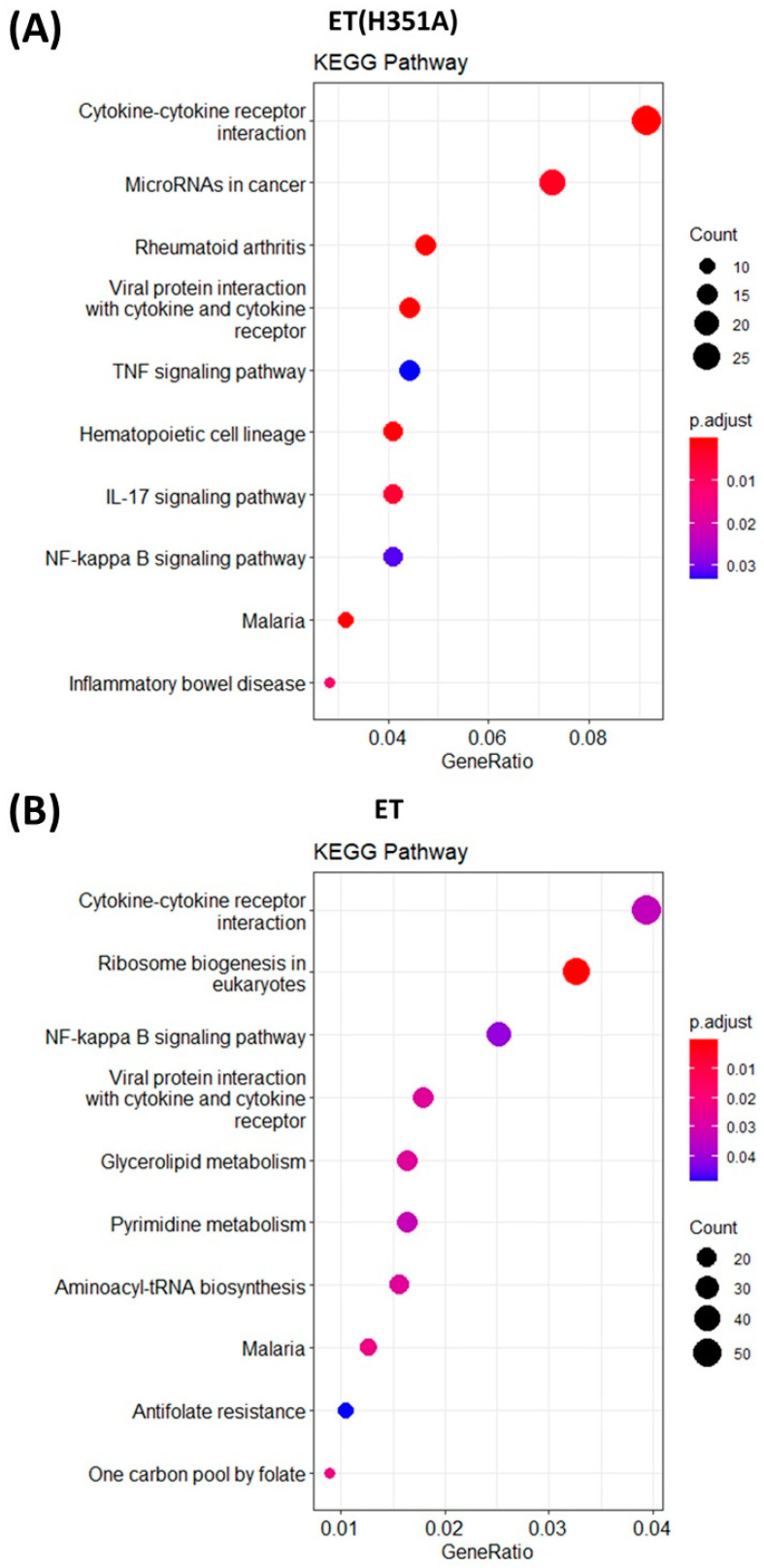
Pathways affected by ET or ET(H351A) challenge. The list of differentially expressed genes (DEGs) (|log_2_ FC| > 1; *p* < 0.05) for each group was subjected to pathway enrichment analysis using the KEGG database. We determined the top 10 modulated pathways in the transcriptome of ET(H351A)- (**A**) or ET- (**B**) challenged macrophages. Pathways were graphed according to the GeneRatio, defined by the ratio between the total number of DEGs and number of genes that belong to a pathway. The dot size is correlated with the number of genes that belong to a pathway. Dots are colored according to the adjusted *p*-values (*p*-adjust) from blue (higher *p*-value) to red (lower *p*-value).

**Figure 6 toxins-15-00139-f006:**
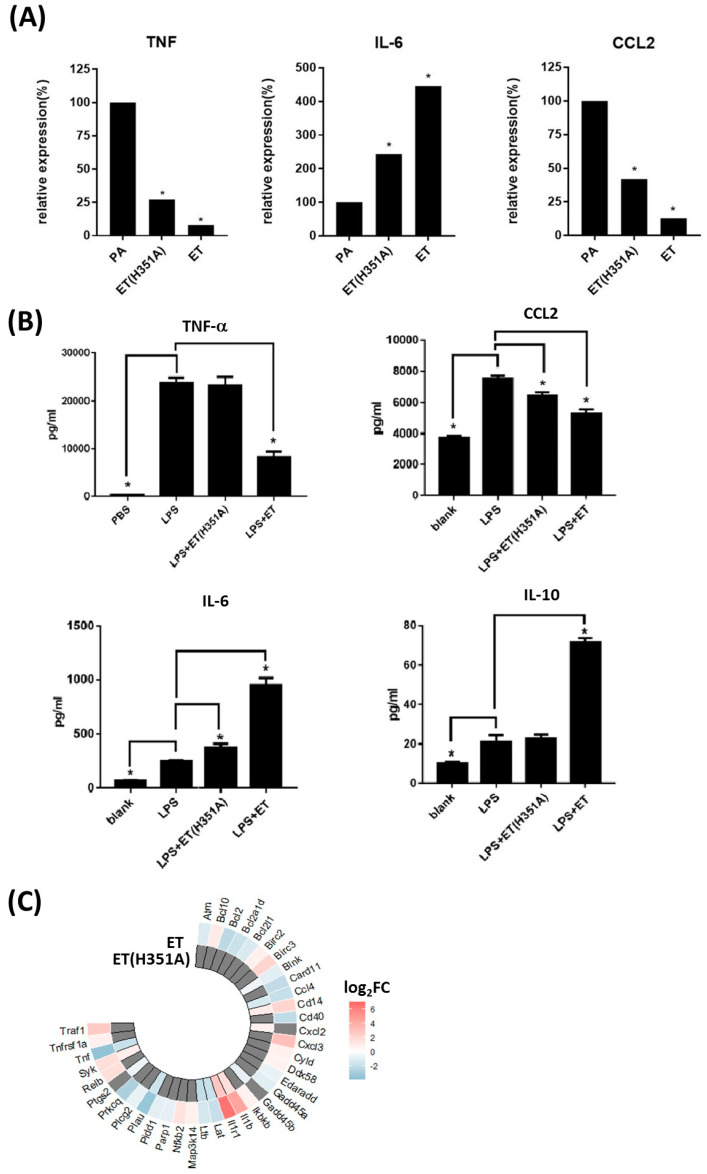
ET(H351A)- and ET-induced cytokine expression changes. Three cytokine genes were chosen to validate the RNAseq data by cytokine secretion analysis. (**A**) From the RNAseq data, the fragments per kilobase of transcript per million mapped reads (FPKM) of TNF, IL-6, and CCL2 were plotted as means. (**B**) The levels of TNF-α, CCL-2, IL-6, and IL-10 produced by the RAW264.7 cells treated with PBS, LPS, LPS + ET(H351A), or LPS + ET for 3 h. Results represent the mean ± SD of three independent experiments. * *p* < 0.05. (**C**) Heatmap of the DEGs in the regulation of the NF-κB signaling pathway. Red and blue represent up-regulation and down-regulation, respectively, of the mRNAs.

**Figure 7 toxins-15-00139-f007:**
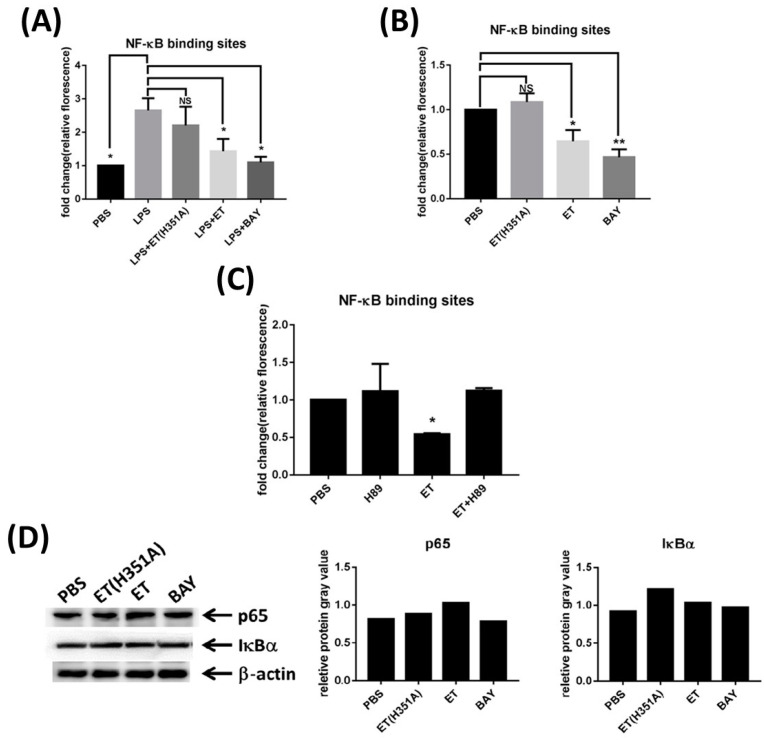
ET downregulates the transcriptional activity of NF-κB and phosphorylation of p65. (**A**,**B**) Relative luciferase expression in RAW264.7 cells after transfection with the NF-κB activation reporter plasmid and the control plasmid, followed by 3 h of treatment with PBS, 100 ng/mL ET (H351A), 100 ng/mL ET, or 10 μM BAY 11-7082 (BAY) in the (**A**) presence or (**B**) absence of LPS. (**C**) Relative luciferase expression in RAW264.7 cells after transfection with the NF-κB activation reporter plasmid and the control plasmid, followed by 3 h treatment with PBS, 10 μM H89, 100 ng/mL ET, or both H89 and ET. (**D**,**E**) Protein expression of (**D**) IκBα and p65, as well as (**E**) p65 phosphorylation in RWA264.7 cells treated with PBS, 100 ng/mL ET (H351A), 100 ng/mL ET, or 10 μM BAY for 3 h. Shown are representative Western blots, and the relative protein expression was normalized against β-actin and quantified by densitometry. * *p* < 0.05, ** *p* < 0.01.

## Data Availability

The data presented in this study are available in this article and Appendix A. The datasets generated for this study can be found in the SRA(The Sequence Read Archive), accession PRJNA797714.

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
