# Peer review of "Whole-Transcriptome Analysis Highlights Adenylyl Cyclase Toxins-Derived Modulation of NF-κB and ERK1/2 Pathways in Macrophages"

_toxins, 2023, doi:10.3390/toxins15020139_

Round 1

Reviewer 1 Report

In this manuscript, the authors systematically analyzed the mouse macrophage cell line treated with the anthrax edema toxin at transcriptomic level. ET induced a subset of differentially expressed genes that involve in multiple cellular functions, such as ERK1/2 and NF-kB signaling pathways. The experiments, especially the global RNA profiling, were designed and performed well. The manuscript was organized well as well. Although the in vivo evidence is missing, I believe the concept itself is interesting and the major findings could benefit the field. To move the manuscript forward to publishable level, a couple of minor issues need to be addressed:

1.      Species, such as B. anthracis (line 41) and E. coli (line 468), need to be italic.

2.      It is good to compare the differentially expressed genes between wildtype and mutant ETs.

3.      Did the authors remove the endotoxin from recombinantly purified proteins? The better control would be 200 ng/mL PA plus 100 ng/mL boiled ET to normalize the protein amount.

Author Response

Point 1: Species, such as B. anthracis (line 41) and E. coli (line 468), need to be italic.

Response 1: I have corrected the species names into italic.

Point 2:  It is good to compare the differentially expressed genes between wildtype and mutant ETs.

Response 2:

The different biology effects induced by wildtype and mutant ETs must be of important scientific value, but not the main purpose of the present study. To demonstrate influence of ET(especially its adenylate cyclase activity) on macrophages, the present study focus more on the similarities between the differentially expressed genes(DEGs) induced by ET and ET(H351A) treatments.

    However, when we pay close attention to the differences in KEGG pathway enrichment of the DEGs induced by ET and ET(H351A) treatments, it is interesting to find that several pathways involved in cell proliferation were enriched in ET treated group but not ET(H351A) treated group.

   The following contents have been added into the manuscript:

   Moreover, several pathways involved in cell proliferation (ribosome biogenesis in eukaryotes, aminoacyl-tRNA biosynthesis, pyrimidine metabolism, one carbon pool by folate) were just enriched in ET treated group but not ET(H351A) treated group.

In eukaryotes, cAMP synthesis is canonically triggered via G protein-coupled receptor (GPCR)-mediated activation of endogenous transmembrane ACs. The functional diversity of cAMP signaling is tightly regulated by intracellular cAMP gradients and microdomains[47]. The destruction of cAMP compartmentation in normal cells increases cell proliferation and induces cell transformation[48]. ET, as an exogenous AC, intensively elevates the cAMP level in macorphages independent of GPCR. The cAMP molecules induced by ET are very likely distributed randomly within the cell. That may be the reason why ET treatment affected several cell proliferation relevating pathways.

Point 3: Did the authors remove the endotoxin from recombinantly purified proteins? The better control would be 200 ng/mL PA plus 100 ng/mL boiled ET to normalize the protein amount.

Response 3: The endotoxin is removed from all the proteins used in this study. The detailed method had been described in the following references:

  1. Zhao, T.; Zhao, X.;  Liu, J.;  Meng, Y.;  Feng, Y.;  Fang, T.;  Zhang, J.;  Yang, X.;  Li, J.;  Xu, J.; Chen, W., Diminished but Not Abolished Effect of Two His351 Mutants of Anthrax Edema Factor in a Murine Model. Toxins (Basel) 2016, 8 (2), 35.
  2. Xu, J. J.; Dong, D. Y.;  Song, X. H.;  Ge, M.;  Li, G. L.;  Fu, L.;  Zhuang, H. L.; Chen, W., [Expression, purification and characterization of the recombinant anthrax protective antigen]. Sheng wu gong cheng xue bao = Chinese journal of biotechnology 2004, 20 (5), 652-5.

It is a good suggestion to use 200 ng/mL PA plus 100 ng/mL boiled ET as control, we will take this advice in our further study.

Reviewer 2 Report

This is an interesting study that investigates the impact of Adenylyl Cyclase Toxins on macrophage polarization and the NF-kB or ERK-Pathway. The data is clearly presented, not over-interpreted and the methods used are state-of-the-art.

However, minor comments could be addressed before publication of the manuscript:

Interpretation of data in terms of macrophage polarization is rather difficult from the current data, but could be of great interest. Although the stimulate RAW-macrophages produce high levels of IL-10, also IL-6 is elevated. Since the authors` analysed the macrophage transcriptome, it would be interesting to match it with published mouse macrophage polarization phenotype transcriptome data that are eventually published. At least, key macrophage polarization transcription factors could be mapped in the data generated from this study.

These data could shed “more light” on the macrophage phenotype in response to B. anthracis.

Author Response

Point 1: Interpretation of data in terms of macrophage polarization is rather difficult from the current data, but could be of great interest. Although the stimulate RAW-macrophages produce high levels of IL-10, also IL-6 is elevated. Since the authors` analysed the macrophage transcriptome, it would be interesting to match it with published mouse macrophage polarization phenotype transcriptome data that are eventually published. At least, key macrophage polarization transcription factors could be mapped in the data generated from this study.

 Response 1: We tried to match our RAW-macrophages transcriptome data to published mouse macrophage polarization phenotype transcriptome data. But during the process of searching a proper macrophage polarization phenotype, we realized that M1 and M2 subtypes just representing two poles of macrophages heterogeneity. In the real in vivo environment, especially in human body, macrophages differentiate as a continuous spectrum responding to stimulations. Under the situation of cancer, it is very common to find that tumor-associated macrophages show M2-like phenotype which expressing both pro-inflammatory and anti-inflammatory cytokines.

Moreover, macrophages differentiation is not only regulated at the transcription level, but also at the metabolism and post-transcriptional modification level.

So, the definition of the macrophage phenotype in response to B. anthracis may take more effort which is beyond the scope of the present work. We have found some interesting changes in cell metabolism phenotype and protein acetylation induced by B. anthracis. A comprehensive analysis will be taken in our further work.

Reviewer 3 Report

The authors analyze the effect of cAMP production by B.anthracis edema toxin in macrophages. They show that specific transcriptional changes lead to modulation of multiple macrophage functions. PKA seems involved in this signal pathway.

The data presented are interesting, clear and convincing. The discussion needs to discuss the following questions:

Is there a preference of ET injection over LT injection into macrophages?

Is there anything known about the connection of PKA activation and gene synthesis or can you suggest a pathway explaining PKA induced changes in gene expression?

Minor:

line 54: do you mean "inhibited" instead of "degraded"?

line 95: "inhibitory" instead of "inhibition"?

line 96: LPS-induced

line 100: H89 is definitely not a PKA agonist. It is an inhibitor.

line 106: "is involved" instead of "involves"?

line 176: processes

line 229: ERK1/2 insted of REK1/2

Author Response

Point 1:Is there a preference of ET injection over LT injection into macrophages?

Response 1:To date, there is no evidence showing the preference of macrophages to ET or LT. ET and LT share the same mechanism to enter macrophages, so theoretically the chances for both toxins to enter macrophages should be equal. But in vitro study showed that a low level of LT could induce toxin resistance in macrophages, suggesting that cells may adapt so as to tolerate toxic doses of LT. Moreover the cumulative biological effects of ET leading to formation of extensive edema combined with the release of capsular material by the bacteria may also decrease the ability of LT to diffuse extracellularly and interact with its cell receptors, thus further decreasing its toxic effects. That may be the reason why ET exerted an overall predominant effect during the initial stages of B. anthracis inhalational infection.

Point 2: Is there anything known about the connection of PKA activation and gene synthesis or can you suggest a pathway explaining PKA induced changes in gene expression?

Response 2: In the present study, inhibition of phosphorylation of p65 S536 and down-regulating of NF-κB activity by ET was found to be the core mechanism of ET induced changes in gene expression.  Phosphorylation of p65 S536 enhances transactivation by recruitment of histone acetyltransferases. Several kinases were reported to S536 residue of p65 could be phosphorylated by several kinases, including IKKα, IKKβ, Akt, TANK-binding kinase1 (TBK1), IKKε and so on. Among these kinases, non-canonical IκB kinases TBK1 and IKKε activity was reported to be inhibited by cAMP increasement and PKA activation. This may be the possible mechanism of PKA inhibiting NF-κB transcription activity and induced changes in gene expression. 

Point 3: line 54: do you mean "inhibited" instead of "degraded"?

Response 3: There should be “degraded by”, “by” have been added in the manuscript.

Point 4: line 95: "inhibitory" instead of "inhibition"?

Response 4: The "inhibition" have been corrected into “inhibitory”.

Point 5: line 96: LPS-induced

Response 5: A “-“ have been added between “LPS” and “induced”.

Point 6: line 100: H89 is definitely not a PKA agonist. It is an inhibitor.

Response 6:The " agonist " have been corrected into “inhibitor”.

Point 7: line 106: "is involved" instead of "involves"?

Response 7: The " involves " have been corrected into “is involved”.

Point 8: line 176: processes

Response 8: There is a "processes" in line 176.

Point 9: line 229: ERK1/2 insted of REK1/2

Response 9: The " REK1/2 " have been corrected into “ERK1/2”.